# Stigma Toward Severe Mental Illness Among Healthcare Students: A Qualitative Study

**DOI:** 10.3390/ijerph22030333

**Published:** 2025-02-24

**Authors:** Ana Isabel Masedo Gutiérrez, Geraldine Cedeño Martinez

**Affiliations:** 1Department of Personality, Evaluation and Psychological Treatment, University of Malaga, 29071 Malaga, Spain; 2Master Program of General Health Psychology, University of Malaga, 29071 Malaga, Spain; geraldinemartinez@uma.es

**Keywords:** severe mental illness, stigma, healthcare students, qualitative study, contact experiences, ideals, coercive measures

## Abstract

Background: Individuals diagnosed with Severe Mental Illness (SMI) often perceive negative attitudes from health professionals, acting as a barrier to effective treatment. The present study explores healthcare students’ attitudes toward SMI to identify potential areas of stigma. Methods: A descriptive analytical qualitative approach was employed. Twenty-seven students from six different healthcare programs at the University of Malaga participated in semi-structured interviews, which were subsequently analyzed using thematic content analysis with the assistance of NVivo 12 software. Results: Three main themes emerged from the data: ideals, experiences with SMI, and views on hospitalization and coercive measures. Students acknowledge the importance of empathy but often feel unprepared due to limited knowledge, leading to avoidance and negative interactions. Although stigmatizing beliefs regarding dangerousness and inabilities persist, students generally reject segregation and advocate for equal rights. Positive contact experiences with mental illness can modify negative attitudes and enhance empathy, particularly among individuals with their own personal experiences. Participants generally oppose coercive measures, except for safety concerns and lack of illness awareness, advocating for alternatives to preserve autonomy. Conclusions: The study highlights that insecurity and feeling unprepared could be linked to stigmatizing and negative experiences with individuals with SMI. Therefore, we underline the need for the anti-stigma education of healthcare students focusing on contact experiences and promoting confidence in their knowledge and skills.

## 1. Introduction

Severe Mental Illnesses (SMIs) refer to those characterized by chronicity, prolonged duration, and significant impairment in the individuals’ areas of functioning. Generally, the clinical criteria are associated with psychotic disorders, major affective disorders, and personality disorders. In addition to clinical symptoms, SMIs involve an increased risk of substance abuse, suicide, medical problems, unemployment, discrimination, and stigma [1].

Negative conceptions of these disorders have been perpetuated in society due to the initial approach being based on isolation, until a community-based approach was embraced, advocating for integration within their social context [1]. Despite this shift, misconceptions and stigma persist across cognitive, emotional, and behavioral aspects, including perceptions of danger, incurability, irresponsibility, apprehension, attribution of blame, prejudgment, avoidance, and discriminatory actions [1,2,3].

Although evidence suggests that healthcare professionals tend to exhibit fewer stigmatizing dynamics in comparison to the general population, they are still prevalent and limit the approach, recovery, and quality of life of affected individuals [3,4,5]. Findings indicate that some health professionals may display even more public stigma than the general population [6]. As a consequence, patients often feel discriminated against and rejected by healthcare staff when seeking help for both physical and mental problems, reporting poor treatment, over-medicalization, and disregard for their opinions [7,8,9,10,11,12]. In addition, these encounters can directly impact their self-esteem and willingness to seek help, contributing to social isolation and acting as a barrier to getting treatment [12].

Stigma reduction among professionals may be achieved through targeted mental health education early in their formative years [13,14,15], promoting empathy and improving their approach. However, research suggests that current programs may not significantly reduce stigma [16], signaling a need for deeper examination of the underlying factors. Given the predominant focus on quantitative measures in current research, a qualitative approach could provide valuable insights to address current gaps, as it allows for a deeper understanding of social representations with regard to a specific problem [17,18].

Existing qualitative studies have deepened our understanding of stigma. For instance, a study that reviewed qualitative research found that insufficient mental health training impacts healthcare professionals’ confidence, resulting in negative attitudes and increased social distance toward individuals with mental illness [19]. Similarly, other studies highlight that mental health education and direct contact can reduce stigma [20,21]. However, much of the current research tends to narrow its focus on specific groups or disorders. Despite providing findings across various healthcare professionals and mental illnesses, Riffel and Chen’s qualitative review (2020) still emphasizes the ongoing need for further research, particularly concerning the impact of contact experiences [19].

Thus, the aim of the present study is to explore healthcare students’ attitudes and perceptions regarding SMI in order to identify possible patterns, recurring themes, the impact of previous contact, and educational gaps. The results could lead to valuable insights and pave the way for targeted intervention programs to mitigate stigma, and consequently, improve the quality of life of people with SMI.

## 2. Methods

In the present study, a descriptive analytical design was employed based on a thematic analysis through Nvivo12 version Release 14-23-1 (38) software [22], which allows one to organize patterns, nodes, and subnodes that emerge as relevant and prepare them for analysis [23]. The protocol was approved by the University of Malaga’s Experimentation Ethics Committee (CEUMA) with the registration code 40-2023-H (date of approval: 13 June 2023).

### 2.1. Participants and Recruitment

Students were recruited voluntarily from different healthcare disciplines at the University of Malaga via email and class announcements. To promote a diverse sample, variables such as specific degree programs, academic years (first or second cycle), gender, and personal contact with SMI were taken into consideration. Efforts were made to obtain a balanced distribution of participants across these variables, ensuring a representative sample.

The inclusion criteria were students within the first and last year of a healthcare program, and the only exclusion criterion was if the student had received specific education regarding mental health stigma. There was not a sample size previously defined other than having a minimum of two interviews per healthcare program. The interviews were conducted until theoretical saturation was reached, which is the point at which additional data do not add any more insight into the new theory [24,25].

### 2.2. Data Collection

A thematic script inspired by Rodriguez-Almagro’s study (2019) was developed as a basis for the semi structured interview and the first stage of coding [20] (see Appendix A). The script included four dimensions, knowledge, attitudes, social distance, and behavior, each explored through open-ended questions. Participants were given a brief explanation of the purpose of the study as well as a definition of severe mental health disorders prior to signing the consent form.

The interview was divided into three sections: twenty-one questions regarding sociodemographic data, twenty-two questions based on the dimensions described in the thematic script, and two optional questions. The sociodemographic data were obtained with the aim of organizing the results and finding solutions to the possible level of stigma within the sample. The core of the interview consisted of open-ended questions directed toward knowledge on mental health issues, perceived attitudes, stances with regard to social distance, and behavioral responses. Lastly, the two additional questions were more personal and therefore optional, as they were directed toward the participants’ personal experience with mental health stigma.

### 2.3. Data Analysis

The transcribed interview data were organized using Nvivo12 version Release 14-23-1 (38) software [22]. A content analysis technique was applied to examine and identify relevant and recurring themes in participants’ responses regarding SMI. The thematic analysis followed the guidelines proposed by Castro et al. (2022) [23] and it was conducted in four phases that are not linear, but rather overlapped each other.

In the first phase, researchers familiarized themselves with the data by listening to recordings and transcribing them, assigning pseudonyms to each participant to ensure anonymity. Initial readings focused on identifying units of meaning, referring to citations of phrases with greater relevance to stigma, which were subsequently coded.

During the second phase, nodes or central categories were generated. The main researcher created an initial categorization of the units of meaning into nodes, forming a hierarchical thematic matrix, using both deductive and inductive approaches. The deductive approach followed the coding tree by Rodriguez-Almagro and colleagues [20] and extensively considered the previous literature and theories, while the inductive approach allowed the identification of initial themes. Nodes were reapplied to the transcriptions, allowing for the identification of new subcategories that emerged as specific themes within the central category.

The third phase involved a thorough review and refinement of the units of meaning coded into central categories to identify new subnodes. Special consideration was given to ensure each node was broader than a subnode.

In the fourth and final phase, the first version of the thematic matrix was reviewed, and the identified categories and subcategories were defined and provided to two additional researchers to unify codes. Each researcher coded two additional transcriptions, ensuring agreement on categories and a thorough, consistent analysis.

The resulting codebook was applied to three new transcriptions to increase reliability. Theoretical saturation was reached when no new categories emerged and subnodes with few references were expanded, requiring at least four references per subnode. A highly discriminative strategy ensured each reference was included in only one node, creating a new subnode if necessary, based on the final number of references. A detailed evaluation of the hierarchy of nodes and subnodes took place to ensure the denominations transmitted the results accurately.

## 3. Results

A total of 27 students from six different healthcare programs at the University of Malaga were interviewed, and their demographic data are represented in Table 1. The programs represent both biomedical (medicine, nursing, and physiotherapy) and psychosocial disciplines (psychology, social work, and occupational therapy). Most of the participants were female (59%) and they were similarly distributed across the first two years (41%) and last two years (59%). Approximately 40% of the participants have had previous contact with SMIs. Of the 18 participants, 9 of them reported mental health concerns themselves. For more specific details with regard to healthcare programs and participant pseudonyms, see Appendix B.

Upon analyzing the data collected in the interviews, three recurring themes were identified: ideals, contact experiences with SMIs, and opinions regarding coercive measures (see Figure 1). These categories emerged as a representation of the relationship between stigma, ideals, and experiences of healthcare students. Within each node, there are subnodes that provide a deeper insight into participants’ perspectives. To enhance clarity and comprehension, excerpts from the interviews are included, referencing the participants’ designated pseudonyms.

### 3.1. Ideals

The participants present different ideals with regard to mental health, including the attitudes every professional should have, what they wish to learn, knowledge and concepts regarding people with SMI, their rights and duties, and lastly, their perspective on segregation.

#### 3.1.1. Professional Attitudes—“Open Mind Without Prejudices”

Regarding perspectives on professional attitudes, the participants emphasize the need to maintain an open-minded and positive approach toward people with SMIs. They particularly highlight the importance of active listening without prejudices, as well as showing interest in understanding the patient. Participants believe that for this to occur, the professional must have empathy, an ability they associate with having prior encounters with mental health issues.

When discussing underlying motivations for providing an adequate treatment toward patients, Artaud explains that “we are all going to go through something like that”, and Gibson believes that it might be harder to understand the patient because “many of us have never been through a similar situation”. On the other hand, Gomez suggests that the level of empathy does not depend on prior contact, but rather on the professional roles, as he states that it is “not the same being a psychologist, who is capable of placing themselves in their place, versus other professions where the patient is not as ‘important’”.

Furthermore, there seems to be different perspectives as to what the role of the professional should be and/or where they would ideally place the focus on. Millet and Woolf, who are in the biomedical field, highlight the importance of “curing (…) and giving out information”, whereas Nash and Pessoa, pursuing psychosocial studies, emphasize the importance of the contact with the person and placing effort into creating a “trustworthy environment”.

#### 3.1.2. What They Wish to Learn—“They Should Teach Us More How to Treat with These People”

The majority of participants express a desire for more practical knowledge in their studies, as they perceive deficiency in this area and indicate a wish to improve their communication and intervention skills toward people with SMIs. Schuman believes it is important because “it’s not the same talking to someone who has a trauma, than for example someone with agoraphobia” and Felipe V. affirms the importance relies on the fact that even if you are not a patient’s main doctor, you “still need to have a certain treat towards that person”.

Participants acknowledge feelings of uncertainty and a lack of confidence in their professional abilities, emphasizing the importance of embracing a practical and gradual learning process. Pollock affirms “we graduate with a little bit of panic (…) of being incompetent, of not having any idea on anything, of having studied the theory but then not knowing how to behave”.

Most participants believe this learning gap could be addressed by incorporating practical cases, speeches delivered by people with SMIs, and “seminars that make us see things much more humanly”, and be “less frivolous and more sensitized” (Carey). Two of the participants mention interest in gaining additional theoretical understanding regarding the origins of mental disorders and believe that “they should add more psychology classes” (Felipe V.).

#### 3.1.3. Rights and Duties—“Same Rights but Not Duties Since Cognitively, They Function Differently”

Participants express varied perspectives on whether individuals with SMIs should have the same rights and duties as the rest of the population; whilst most of them confidently affirm that they should have the same rights, doubts arise with regard to duties.

Gaga and Artaud argue that individuals with SMIs deserve the same rights but not the same duties since they “would not have the necessary mental capacities”. Simone believes there should be flexibility within the duties when “going through a rough patch”. With regard to both rights and duties, Haendel suggests that, since mental faculty is affected, “it depends on the level of severity of the patient”, and Millet seems to agree, adding that “there would be a lot of psychological tests needed to remove a person’s power of decision making (rights)”.

Lastly, although participants acknowledge the importance of equal rights for individuals with SMIs, some suggest contradictory stances. For instance, Beret proposes for them to be “isolated or guarded alone in a room for a few days until they feel better and stop having those types of actions”, while Artaud remarks on the importance of caution and sensitivity in interactions. Moreover, Cher and Carey display uncertainty regarding the topic, answering with a doubtful tone and following it up with a “right?”.

#### 3.1.4. General Knowledge—“I Consider My Knowledge Is Limited”

Participants disclose limited knowledge with regard to severe mental disorders, including brief definitions and treatment approaches. They tend to describe SMIs as disabling disorders that significantly impair people’s lives. Participants recognize that the non-physical nature of SMIs hinders our awareness of them in society. Some display avoidance of the topic due to its perceived complexity and Poe states “I may even have prejudices, I don’t know”.

Certain ideas emerge regarding the abilities of people with SMIs, such as “they are like normal people, the only thing is they have a slight deficiency” (Baret) and the belief that they have “an inability to develop as a person” (Woolf). Schuman states that they are “people who have always been diagnosed or suffered from the disorder”. Concerning her personal experiences, Simone suggests that people with SMIs are characterized by a “more negative perception about life” that may lead them to a decrease in personal satisfaction.

With regard to the treatment, Pollock supports the idea that “we do not need to make a diagnosis to do our jobs” but instead emphasizes the importance of understanding individuals’ challenges. Some participants remark on the necessity of complementing the pharmacological treatment with therapy, stating that without it, you are “postponing that disorder because it is not solving anything…”. Other participants highlight factors such as camaraderie and engagement in activities as contributing factors in improvement.

#### 3.1.5. Specific Knowledge—“Haven’t Been Taught How to Treat Them”

With regard to specific knowledge, participants allude to familiarity with the DSM 5 Criteria, risk signs, and some intervention strategies for SMIs. However, they believe there is limited knowledge and “have a feeling that although we have seen a little bit of everything, we have not dedicated an extra that might be pertinent” (Pollock). Munch has broader knowledge on pharmacology but critiques the approach taught in psychiatry: “we haven’t been taught how to talk to them, how to treat others, how to put empathy into it…”.

Nonetheless, some participants convey confidence in their knowledge and ability to integrate diagnostic criteria with intervention techniques. Schuman explains that a psychological intervention is “necessary, as it’s very hard that one can solve it by themselves, and it may be accompanied by a pharmacological treatment in the more complex cases”. Simone notices an increase in destigmatization, as she has learnt that disorders are “more common (…) and not as bad or as taboo” and they have a solution, while Van Gogh agrees and stresses the need for early intervention. Lastly, Nietzche acknowledges the variability within different disorders depending on individual and contextual factors.

#### 3.1.6. No to Segregation—“It Does Not Make Sense to Isolate Them”

Most participants advocate for integration, rejecting the idea of isolation as some have expressed that it would be counterproductive. Artaud states that isolation “would mean not improving in other areas of your life (…) and it would make them feel discriminated against”. In addition, he advocates for specific mental health programs to be implemented in order to facilitate social inclusion. Similarly, Van Gogh emphasizes the need for integration as it is “necessary for visibility and for everyone to see that it does not matter what you have if you have a treatment”.

However, a few participants mention exceptions to this perspective. For instance, Hendrick and Cher suggest that individuals with a problematic background may require segregation, regardless of whether the person has a mental disorder or not, while other participants mention cases where temporary isolation might be necessary for safety reasons, both toward society and themselves.

Despite the exceptions, participants tend to reject the idea of segregation, emphasizing that it perpetuates fear and stigma surrounding SMIs. For example, Lovato believes this association is unfair, as “you might find the most pacific person in the world or the most assertive and that person might have a SMI”.

### 3.2. Contact Experiences with Severe Mental Illness

Participants unveil a variety of experiences, ranging from positive insights to complicated situations as well as their own experiences.

#### 3.2.1. Positive Experiences—“They Are Good People”

Participants acknowledge the presence of prejudices prior to establishing direct contact with people with SMIs but often change their perspective after interacting with them, leading to a more positive experience. Those who have only had professional interactions describe the individuals with SMIs as nice and pleasant, whereas those who have had personal contact elaborate upon their experiences.

Regardless, participants generally express surprise at the realization that people with SMIs “are like any other person” after establishing contact. In particular, Poe describes feeling scared prior to the contact and thinking “I found a super big surprise (…) Amazing honestly, I developed a lot of affection towards him”. Similarly, Cher shares she has always had a slight fear toward mental illness, but has a colleague with OCD, and “the more I am with him, it does not matter, he is a normal person”. Even participants whose experiences are more indirect agree, such as Gibson, whose aunt’s neighbor “is schizophrenic and they coexist in a community, so I don’t think it’s a big issue”

#### 3.2.2. Impactful Experiences—“Shocking”

Several participants describe their experience with mental illness as shocking, such as the case of Johnston, who affirms “I don’t know how to explain it, with the medication he was ok and I had conversations with him like a normal person, but I have also seen heavy episodes, very heavy episodes”. Similarly, Simone notes that “it opened my eyes because I couldn’t see it from the outside”.

#### 3.2.3. Hard to Deal with—“It’s Very Complicated”

Some participants hold different beliefs and prejudices that leads them to find interactions with people with SMI challenging, such as the case of Gaga, who expresses “it’s scary, like how are you going to deal with that person”, while Hopkins places his boundary: “I wouldn’t have a problem with maintaining a friendship, but anything more than that, I think I wouldn’t be ready to put up with it”. Other participants would embrace a cautious approach, treating them “as a normal person, yet always being careful, right? (…) because the reality is they can have a ‘tostón mental’” (referring to mental disturbance), and being “careful because you don’t know how they’re going to react”.

Some participants have had more personal and direct negative contact, expressing to have observed harmful actions. For instance, Lovato describes people with SMIs as being “so afraid and have so much anxiety that they manipulate a lot and do emotional blackmail”, and Pollock shares this sentiment, remarking on the necessity to “put limits to the relationship because sometimes it is simply not sustainable, and it doesn’t do you good”.

#### 3.2.4. Own Experiences with Mental Illness—“I Felt the Stigma”

During the interview, participants recalled their personal encounters with mental illness and shared details about the stigma they perceived and factors that helped them. Haendel reflects on the importance of acknowledging the problem because “sometimes you can have symptoms and not know what is going on and when they tell you, look, this is what’s happening to you, you kind of assimilate it and say, oh, ok”. Conversely, Hopkins experienced relief after receiving a diagnosis as it helped “understand my behavior and feelings. It cleared things up”.

Despite the support received by some of their friends, some participants still struggled with stigma on some level. Pessoa was skeptical about disclosing his struggles and approaching a psychologist due to the social stigma; however, after positive feedback from his context, it was “a relief because (…) it took away my own stigma on mental health”. For others, as is the case of Schuman, the stigma becomes more apparent over time and especially within their personal relationships, as people seem to become distant, act in “shock and pitiful” and “scared, like they didn’t know how to act around me”.

Participants tend to withhold their mental health issues from their families, perceiving them as less understanding and supportive. One participant expresses “they are not very -talkative about feelings-, they hide it more, they don’t want me saying it around (…) they see it as a failure” (Schuman).

### 3.3. Hospitalization and Coercive Measures

Most of the participants believe it depends on the case, but that coercive measures should be avoided unless it is completely necessary for safety reasons.

#### 3.3.1. With Their Consent—“As a General Rule, Always with Their Consent”

Participants acknowledge the complexity of the situation and insist on the importance of case-by-case consideration. Regardless, they express that as a base, admission should only be done if the person agrees and “that’s what the informed consent is there for” (Barret). Hemingway believes that coercion should be avoided unless the “treatment benefits them” and the patient is informed and agrees. Other participants propose a gradient approach, prioritizing the patient’s consent first, but if that is not possible, involving the family or legal guardians.

#### 3.3.2. For Their Own Safety—“Only if It Is Strictly Necessary”

Participants recognize forced admissions might be necessary when the patient’s health is in jeopardy. Some of them express discomfort and find some of the measurements “a little aggressive, but then I also understand the situation from the sanitary point of view, so I do see it right” (Poe). Regardless, participants seem to understand the long-term benefits and recognize that “professionals always try to give a good treatment even if it means conflicting situations at some points (…), where they have to impose on their routines to create new habits, but always thinking about the patient’s autonomy”.

#### 3.3.3. If Dangerous—“Those Types of People Could Risk Others’ Lives”

Participants often express that involuntary admission and coercive measures should be avoided unless individuals with SMIs pose a threat to themselves or others. Participants express dissonance as “a part of me wants to say no to involuntary admissions, but another part of me wants to say yes, because deep down there are people in manic episodes who can put their lives and other people’s lives in danger” (Cher). In addition, he suggests other measurements such as “calling the police (…) I think that would be logical right? I am not going into anything that can put my physical integrity in danger”.

Similarly, Johnston displays dissonance when defending that it “depends on the type of mental illness (…) because they are people that… well it’s that I don’t like to generalize, but yeah, I know that there are cases of some disorders and some people who might or might not have schizophrenia, but in some cases, which is also bad, we tend to think that anyone with schizophrenia is aggressive”.

Negative experiences have influenced opinions, as one participant cites that “many classmates have had injuries given by these patients” but also reflects that the aggression might be a consequence of coercive measurements (Woolf). This participant remarks that “we should be careful because you don’t know how they’re going to react”.

#### 3.3.4. Never—“It’s Invading Their Autonomy”

While some participants debate the ethical implications of involuntary treatments and hospitalization, others firmly believe that such measurements should never be employed as they violate their autonomy and “infringe upon human rights” (Van Gogh). Pessoa believes that forcing patients into treatments is “precisely what incites more medicalization and treating them like ‘crazy’”. Furthermore, others state that “these disorders do not even need to imply that the person is dangerous, so they are not going to necessarily do anything to you (…) as if they were a criminal…” (Spears).

Participants question the necessity of such actions and question “to what extent are there actually no alternatives to that?”, and in the same vein, others consider that “people who treat these types of disorders know a lot about them and have the necessary skills, which don’t need to be inadequate”. Simone believes that, anyway, “if the person does not consent, the treatment will most likely not work, and you can’t force them”.

#### 3.3.5. If They Lack Awareness—“They Don’t Have Illness Awareness”

In situations where they lack awareness of their condition, some participants find it necessary to intervene, even if it requires coercion. Hendrix argues that in such cases, “to improve, it needs to be a little bit forced”. Others highlight that if the person is not lucid, “they can’t make their own decisions because they are not ok psychologically, so a family member should help, right?”.

#### 3.3.6. If the Professional Recommends It—“They’re the Experts”

Some participants believe that involuntary admission could be carried out when the professionals recommend it, since “they are the experts”. Woolf expresses concern about the frequency of coercive methods employed by professionals as the first option rather than last resort, because “it is the easy way, they take it as a first option, and I don’t like it.”

## 4. Discussion

The present study explores different beliefs and attitudes healthcare students display toward individuals diagnosed with Severe Mental Illnesses. The results indicate a noticeable presence of stigma toward SMIs, manifested mainly as avoidance, fear, and confusion, possibly related to limited knowledge and lack of contact. However, positive attitudes are also present, as students confront some of the traditional beliefs and advocate in favor of people with SMIs, particularly when they have had prior contact experiences.

### 4.1. Ideals Healthcare Students Present Toward SMIs

Concerning professional attitudes, participants emphasize the importance of empathy in their interactions with individuals with SMIs, with previous studies suggesting that empathy and compassion lead to better attitudes and subsequently reduce stigma [26,27]. Moreover, they express concerns about lacking practical knowledge and a desire to learn more relational skills toward patients [9] while also displaying limited theoretical knowledge. Research has found that students often lack confidence in their knowledge and abilities, feeling inadequately equipped [20,21], frequently resulting in feelings of avoidance, fear, and discomfort [28,29] that likely result in negative professional–patient interactions as well as perpetuating stigma [11].

Participants share a consensus on the need for equal rights while adjusting duties based on perceived cognitive limitations and deficiencies among individuals with SMIs. According to previous research, stereotypes frequently include beliefs regarding weakness and general inabilities [30]. However, some participants present contradictory stances such as advocating for equal rights while also proposing temporary isolation until the patient improves. These contradictions may reflect internalized stigmatizing attitudes or beliefs, such as paternalistic attitudes for example, which, according to authors, are prevalent and seem to drive control over decision making on individuals with mental illnesses [6,31].

However, positive beliefs also emerged with regard to the importance of a multidisciplinary approach for effective treatment and a tendency to reject segregation. Based on the ideals identified, it would be beneficial to prioritize the development of empathetic skills in the early stages of their career, strengthen their abilities to communicate effectively with patients [32,33,34], as well as including additional courses to target misconceptions and increase their confidence with regard to their knowledge [14,35].

### 4.2. The Impact of Having Previous Experiences with SMIs

Findings indicate that personal and professional interactions with individuals with SMIs can modify negative attitudes and facilitate positive interactions [3,19,20,36]. However, negative experiences, particularly associated with individuals’ distressing states, may exacerbate stigma, indicating that the nature of the interactions exchanged has an impact on the attitudes displayed [27,37]. For instance, participants who have had intimate relationships with individuals with SMIs express relational difficulties but are willing to maintain friendships with established boundaries [27]. This is in line with research suggesting that previous contact with mental illness is associated with less social distance [38].

Some participants disclosed their own experience with mental illnesses, perceiving stigma and expressing that it is complicated. They manifest hesitation in disclosing their struggles to others due to possible negative reactions [32,39]. While some received adequate support from friends, which could have a positive impact on quality of life [40], others have perceived negative reactions marked by distance and pity, primarily from family members, echoing results from previous investigations [9,41]. For instance, families tend to conceal and avoid the mental illness, which could be driven by feelings of embarrassment or shame, leading to a denial of the disorder [42,43].

One participant acknowledged his internalized self-stigma, which is often prevalent among individuals with SMIs [41,44,45,46], and acts as a barrier to seeking help. Despite this barrier, participants sought help from both professionals and friends, which, as mentioned earlier, sometimes led to negative reactions from family members. Nevertheless, in other cases, it led to a sense of relief through better understanding of their mental health and even reduced their internalized stigma. This is in line with the current literature suggesting that one’s own experiences have an impact on attitudes regarding dangerousness, help, segregation, and avoidance [38,47] and that such experiences enhance empathy, thus leading to less stigmatizing beliefs [21,48].

These findings also explain why in the present study, those who have experienced stigma tend to reject coercive measures and express clearer opinions, compared to those who lack direct experiences. Thus, addressing personal experiences with mental illness contributes to diminishing the perceived distance between themselves and patients.

### 4.3. Perspectives on Involuntary Admission and Coercive Measures

Participants generally concurred that coercive measures should be avoided unless necessary for safety reasons. However, some participants argued against their use altogether, viewing them as an infringement upon human rights and autonomy, and suggesting that they perpetuate stigma. Instead, they advocated for alternative approaches, although specifics were not provided. These findings echo existing research in the mental health field, where professionals express that coercive measures are undesirable but necessary in acute clinical states, and some of them acknowledge their detrimental effect on the individuals’ autonomy, rights, and interpersonal contact [49,50,51].

Participants often justified coercive measures as necessary for providing better care and providing safety [52]. Additionally, a subtheme emerged during the study, with some participants suggesting that involuntary admission might be adequate for individuals lacking awareness of their illness or current clinical state. Although participants refer to having the patients’ best interests in mind, their responses revealed a notable bias toward perceptions of dangerousness and unpredictability, consistent with previous research [49]. For instance, one participant admitted to generalizing, citing the underlying belief that all individuals with schizophrenia are aggressive, thus contributing to the stigmatization of this mental illness that is often linked to dangerousness [53]. Nonetheless, other participants acknowledge that an SMI does not necessarily imply dangerousness.

Throughout the interviews, there was a certain level of dissonance in participants’ responses, indicating potential unconscious biases underlying stigmatizing beliefs or attitudes. Research indicates that specific anti-stigma training may be necessary for healthcare providers to recognize and address their own attitudes [11], thus suggesting that individuals may hold stigmatizing attitudes without being aware of them, which could possibly explain the inconsistencies observed in responses.

Comparing intervention approaches, evidence suggests that strategies based on positive contact with people with SMIs are more effective in the long term in reducing stigma than interventions based solely on theoretical knowledge [37,54]. To guarantee that students obtain such contact, the use of videos, whether real or portrayed by an actor, could be employed as this has been proved to be an effective approach to reduce stigma [33,55,56]. This would help guarantee students receive exposure and indirect contact, enhancing empathy and understanding of SMIs.

### 4.4. Strengths and Limitations

To obtain representative results, data regarding participants’ program, years of study, gender, and prior contact with SMIs were collected. A varied sample consisting of students from six disciplines across the healthcare field was obtained. Although attempts were made to ensure a balanced distribution across programs, it needs to be noted that psychology students were more prevalent than any other program, potentially biasing the results as previous research suggests they tend to present lower levels of stigma in comparison to other disciplines [57].

Additionally, social desirability is often present and can influence participants’ responses, possibly leading to contradictory stances during the interviews and inconsistencies [58]. To address this, rapport was established at the beginning, emphasizing that there were no right or wrong answers, and using open-ended questions to avoid leading responses.

Finally, it is important to note that results of qualitative studies should not be generalized as it is need more subsequent studies with larger samples. If generalization is desired, further quantitative measures would be essential to corroborate these findings. Also, it is important for future qualitative studies to expand the sample, because a small sample size was used in the present study, and this may have limited the findings.

### 4.5. Implications for Future Research and Education

These findings are in line with the current literature, and they contribute to a deeper comprehension of stigma among healthcare students, highlighting the need to devise an effective educational program that targets current gaps in SMIs and prioritizes contact experiences with SMIs and practical and theoretical knowledge to enhance students’ empathy and mitigate stigma. Therefore, we underline the need to promote confidence in knowledge and skills that students are developing in their practical and daily enterprise.

Further research focusing on differences in attitudes and perceptions across healthcare programs may uncover specific challenges to each discipline, allowing for the elaboration and implementation of more targeted educational programs. Additionally, investigating the possible sources of dissonance observed in healthcare students’ responses and examining the extent to which personal experiences impact professionals’ attitudes toward patients with SMIs when self-stigma is present could provide valuable insights.

## 5. Conclusions

The present study reveals important themes regarding healthcare students’ attitudes and perceptions toward individuals with Severe Mental Illnesses (SMIs). The study highlights the impact of contact experiences with mental illness on shaping attitudes and beliefs. Positive contact was found to enhance empathetic and understanding attitudes, whereas negative contact or challenging interactions, especially during acute states, led to caution but did not hinder intimate relationships. Some participants disclosed their own personal mental health experiences, providing valuable insights into perceived stigma, often from family members, but also highlighting the importance of seeking help and support. Furthermore, individuals who have perceived stigma seem to hold a deeper understanding and clarity with regard to SMIs as well as more positive attitudes.

Lastly, most participants opposed coercive measures except in cases of safety concerns or lack of illness awareness, while a few were entirely against it, advocating for alternatives to preserve autonomy and rights. Motives for accepting involuntary admission are linked to dangerousness beliefs, although many participants acknowledge that they are not necessarily related.

The observed dissonance in responses may suggest the presence of unconscious stigma and internalized negative attitudes, highlighting that insecurity and feeling unprepared could be linked to stigmatizing and negative experiences with individuals with SMIs. According to these findings, the training of future healthcare students should prioritize practical knowledge, contact experiences, and promoting confidence in their knowledge and skills as part of anti-stigma-specific programs.

## Figures and Tables

**Figure 1 ijerph-22-00333-f001:**
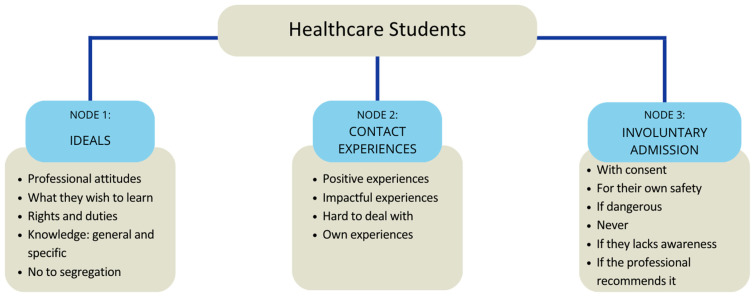
Categories and subcategories that emerged.

**Table 1 ijerph-22-00333-t001:** Demographic data and characteristics of the participants.

Variable	N°
Gender	
Female	16
Male	9
Non-binary	2
Healthcare program	
Biomedical first two years (first cycle)	5
Biomedical last two years (second cycle)	7
Psychosocial first two years (first cycle)	6
Psychosocial last two years (second cycle)	9
Personal contact with Severe Mental Illness	
Yes	12
No	15
Own experiences with mental illness	
Yes	8
None referred	19

Detailed data for each participant can be found in Appendix B.

## Data Availability

Data sharing is not available to this article.

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
