# Peer review of "Stigma Toward Severe Mental Illness Among Healthcare Students: A Qualitative Study"

_ijerph, 2025, doi:10.3390/ijerph22030333_

Round 1
Reviewer 1 Report
Comments and Suggestions for Authors
The study aims to analyze the stigma towards serious mental illnesses among health students. The proposed study is interesting and well structured, especially regarding the results and discussion section. However, in my opinion, the introduction is not sufficiently illustrative of the studies related to your topic and already present in the literature. Although you have cited previous studies, none of them have been explored in depth to support your research hypotheses at a theoretical level. Therefore, I suggest you review the introduction and try to go into more detail on the reasons for your hypotheses.
Furthermore, before proceeding to the description of the participants, I would like to include a paragraph in which the purpose and the starting hypotheses are clearly and in detail, so that the conclusions can be used to discuss the results in light of previous studies and to highlight differences and similarities.
I would also add a section in which you indicate the possible limitations of your study.
I hope that my suggestions can improve the quality of your article.
Author Response
We appreciate so much the Editor´s comments. As we have choosen a qualitative methodology we have a very explorative and constructive point of view. We have not hyphothesis as we follow the Grounded Theory. Then, we have not modified introduction considering that it works for have a general idea of the studies on stigma in healthcare students.
However if the reviewer consider that we need to include some concrete references, we will do it.
Reviewer 2 Report
Comments and Suggestions for Authors
The submitted manuscript is based on a qualitative or descriptive approach, with the primary objective of exploring the attitudes of healthcare students towards individuals diagnosed with severe mental illness. In general, the sections of the manuscript are well-supported, with a coherent structure aligned with the stated objectives.
In the introduction, the research topic is appropriately addressed, highlighting the current issue, the specific objectives, and the purpose of the study. This allows the reader to understand the relevance and context of the research from the outset, establishing a solid introductory framework.
Regarding the method, it is observed that it is suitable for the type of study conducted. The methodology is clearly justified and relevant to addressing the stated objective. Similarly, the results are well-aligned with the proposed method, and their presentation is clear and coherent, which reinforces the validity of the adopted approach.
In the discussion, the expected standards are met by relating the obtained results to previous literature. The implications of the study are also highlighted, and its limitations are initially addressed. However, it is suggested to improve this section by explicitly mentioning the small sample size used, noting how this may have limited the generalizability of the findings. It would also be advisable to include an observation about the need to expand the sample size in future studies to corroborate and strengthen the results obtained.
The conclusion is well-supported, as it adequately summarizes the study’s findings and their relevance in the context of the investigated topic. Nevertheless, it would be appropriate to further emphasize the practical or theoretical implications that could be derived from these findings, in line with the initially stated objectives.
Finally, regarding the references section, it is noted that the journal’s style guidelines have not been followed, particularly regarding the abbreviation of scientific journal names. It is recommended to make the necessary adjustments to comply with the editorial guidelines, ensuring uniformity and professionalism in this aspect.
Specific Aspects to Improve:
- Discussion - Study Limitations: Include an explicit mention of the small sample size used in the study, noting how this may have limited the findings. Additionally, suggest that future studies expand the sample size to validate the results obtained.
- References: Adjust the format of the references to comply with the journal’s style guidelines, including the correct abbreviation of scientific journal names, as indicated in the editorial instructions.
Author Response
Thanks to the reviewer 2 all comments regarding this paper.
Comment 1. “It would be appropriate to further emphasize the practical or theoretical implications that could be derived from these findings, in line with the initially stated objectives.”
We have added a paragraph summarizing important practical and theoretical implications for future research and antistigma education..
Comment 2. Include an explicit mention of the small sample size used in the study, noting how this may have limited the findings. Additionally, suggest that future studies expand the sample size to validate the results obtained.
Thanks very much for the comments. We have added a paragraph mentioning this.
Comment 3. Adjust the format of the references to comply with the journal’s style guidelines, including the correct abbreviation of scientific journal names, as indicated in the editorial instructions.
We have reviewed references that were wrong and it has been corrected.

Reviewer 3 Report
Comments and Suggestions for Authors
This is a topic that explores workplace health for healthcare workers. This has important implications for healthcare management. Comments and suggestions on the content of the manuscript:
1. Summary
It is recommended that the background description get straight to the point. For example: "This study mainly verifies that..."
2. Introduction
While I understand the need to focus on people with SMI, the authors also recommend that such a requirement be incorporated into workplace education for healthcare professionals. But based on the standpoint of equal rights for medical staff, it is important to protect the self-esteem of intensive care patients, and the safety and self-esteem of medical staff are equally important. Because during the care process, medical staff may also suffer verbal and physical abuse, which may affect the individual's physical and mental health in the long run. This will have an impact on medical staff's sense of responsibility, work engagement, etc.
This is important.
3. Methods
Supplement with clear research flow chart and architecture diagram.
4. Results
It is recommended that authors separate the interview content from the discussion content. Clearly present the interview content, as well as the author's reflections and discussions after reading it.
5. Discussion
The content of the discussion needs further reflection to see if it fits the topic and analysis.
6. Other considerations
Authors need to check whether the manuscript layout meets the journal requirements.
Based on the above description, I think this manuscript is meaningful. However, there are still many issues that need to be improved. Therefore, the manuscript needs to be further improved and resubmitted before we can recommend the editor-in-chief to consider further decisions on this manuscript.
good luck,
Author Response
We appreciate so much the Editor´s comments.
Academic Editors’ comments for authors:
1.) To change the structure of the abstract.
As the reviewer suggested, we have move conclusion to the
last sentence. Also, we have modified it as a more conclusive
fragment.
2.) Conclusion too long and more focused
Thanks so much for the comment that contribute to improve
conclusion. We have removed a fragment repeating previous results
and we have added a more focused ad conclusive fragment.

Round 2
Reviewer 1 Report
Comments and Suggestions for Authors
Dear authors,
I am glad that you have modified your paper according to my suggestions.
I still think the introduction needs improvement, but I found the conclusions and future trajectories well described and structured.
Reviewer 3 Report
Comments and Suggestions for Authors
I think the author has addressed the concerns raised in the last comment. I suggest that the next step should be decided based on this revised manuscript.